



**Ecosystem fluxes of carbonyl sulfide in an old-growth forest: temporal dynamics**
**and responses to diffuse radiation and heat waves**
Bharat Rastogi[1], Max Berkelhammer[2], Sonia Wharton[3], Mary E Whelan[4] Frederick C.
Meinzer[5], David Noone[6], and Christopher J. Still[1]
[1] Department of Forest Ecosystems and Society, Oregon State University, OR 97331,
USA
[2] Department of Earth and Environmental Sciences, University of Illinois at Chicago,
Chicago, Illinois, USA
[3] Atmospheric, Earth and Energy Division, Lawrence Livermore National Laboratory,
7000 East Avenue, L-103, Livermore, CA 94550, USA
[4] Carnegie Institution for Science, 260 Panama St., Stanford, CA, USA, 94305
[5] USDA Forest Service, PNW Research Station, Corvallis, OR 97331, USA
[6] College of Earth, Ocean and Atmospheric Sciences, Oregon State University, OR
97331, USA
Corresponding author: Bharat Rastogi (bharat.rastogi@oregonstate.edu)
**Abstract**
Carbonyl sulfide (OCS) has recently emerged as a tracer for terrestrial carbon uptake.
While physiological studies relating OCS fluxes to leaf stomatal dynamics have been
established at leaf and branch scales and incorporated in global carbon cycle models, the
quantity of data from ecosystem-scale field studies remains limited. In this study we
employ established theoretical relationships to infer ecosystem-scale OCS uptake from
concentration measurements. OCS uptake was found to scale with independent
measurements of $CO_2$ fluxes over a 60-m-tall old-growth forest in the Pacific
Northwestern U.S. (45°49′13.76″ N; 121°57′06.88″) at hourly and monthly timescales
across the growing season in 2015. OCS fluxes tracked changes in soil moisture, and
were strongly influenced by the fraction of downwelling diffuse light. Fluxes were also
strongly affected by sequential heat waves during the growing season. Our results bolster
previous evidence that ecosystem OCS uptake is strongly related to stomatal dynamics,
and measuring this gas improves constraints on estimating photosynthetic rates at the
ecosystem scale.

**1. Introduction**
Carbonyl Sulfide (OCS) is the most abundant sulfur gas in the atmosphere, with a mean
atmospheric concentration of ~500 ppt (parts per trillion), and therefore a significant part
of the tropospheric and stratospheric sulfur cycles, with implications for the global
radiation budget and ozone depletion (Johnson et al.,1993; Notholt et al., 2003). The
dominant sink of atmospheric OCS is vegetation (Kesselmeier and Merk, 1993; Kettle et
al., 2002; Montzka et al., 2007 and references therein), through rapid and irreversible
hydrolysis by the ubiquitous enzyme carbonic anhydrase (Protoschill-Krebs, Wilhelm, &





Kesselmeier, 1996; Protoschill-Krebs and Kesselmeier, 1992). Recent advances in
spectroscopic technology have enabled continuous in-situ measurements of OCS on
timescales that are relevant to understanding stomatal function at the leaf-scale (Stimler
et al., 2010a, 2010b), branch scale (Berkelhammer et al., 2014) and the ecosystem scale
(Kooijmans et al., 2017; Wehr et al., 2017). An important distinction between OCS and
$CO_2$ cycling is that there are no reported emissions from actively photosynthesizing
leaves. However, the normalized leaf uptake ratio of OCS:$CO_2$ (LRU; Sandoval-Soto et
al., 2005) is relatively constant at medium to high light levels (Maseyk et al., 2014;
Stimler et al., 2010), making it an excellent proxy for quantifying plant productivity
(GPP; Asaf et al., 2013; Billesbach et al., 2014; Blonquist et al., 2011). On the other
hand, both uptake and emissions of OCS from soils have been identified (Whelan et al.,
2016; Sun et al., 2015; Maseyk et al., 2014; Kesselmier et al., 1999). While ecosystem-
scale measurements of OCS continue to establish links between OCS uptake and GPP in
different ecosystems (for a comprehensive list of ecosystem scale studies readers are
referred to Figure 2 in Whelan et al., 2017), inconsistencies persist. For example, in an
oak-savanna woodland in southern France Belviso et al. (2016) found that OCS exchange
was strongly influenced by photosynthesis during early morning hours, while meaningful
values of LRU could only by calculated for a few days in the early afternoons. Commane
et al. (2015) were unable to explain mid-summer emissions of OCS at a mid-latitude
deciduous forest. Uncertainties highlighted above argue for field-scale measurements of
OCS in a variety of ecosystems, particularly as OCS flux predictions have recently been
incorporated to inform estimates of plant productivity in global carbon cycle models
(Campbell et al., 2017a; Hilton et al., 2017; Launois et al., 2015).
OCS fluxes have not been previously reported for old-growth forests, although a recent
study using flask samples inferred large uptake of OCS in coastal redwood forests in
northern California (Campbell et al., 2017b). Rastogi et al. (in revision) found large
drawdowns in mixing ratios of OCS at an old growth forest in the pacific northwestern
U.S., and significant uptake of this gas by various components of the ecosystem (leaves,
soils, and epiphytes). In this study we report estimates of OCS fluxes from an old-growth
forest and place them in the context of ecosystem carbon and water cycling. Additionally
we investigate the response of $CO_2$, $H_2O$ and OCS fluxes to changes in the fraction of
downwelling diffuse radiation, as well as heat wave events through the growing season.
Technological constraints posed limitations in measuring fast-response OCS fluxes so
instead we combine continuous in-situ measurements of OCS mixing ratios above and
within the canopy with established theoretical equations for OCS uptake (see Berry et al.,
2013; Commane et al., 2015; Seibt et al., 2010) to characterize OCS fluxes using a simple
empirical model and compare them with ecosystem uptake of $CO_2$ from co-located eddy
covariance measurements.
**2. Methods**
2.1. Site Description
Measurements were made at the Wind River Experimental Forest (WR), located within
the Gifford Pinchot National Forest in southwest Washington state, USA (45°49′13.76″
N; 121°57′06.88″; 371 m above sea level). The site is well studied and described in great
detail (Paw U et al., 2004; Shaw et al., 2004; Wharton and Falk, 2016; Winner et al.,





2004). The climate is classified as temperate oceanic with a strong summer drought. The
forest is 478 ha of preserved old-growth evergreen needle-leaf forest, with dominant tree
species of Douglas fir (*Pseudotsuga menziesii*) and Western hemlock (*Tsuga*
*heterophylla*). The tallest Douglas fir trees are between 50 and 60m, while the shade-
tolerant hemlocks are typically between 20-30 m high. Maximum rooting depth is 1–2 m
for the tallest, dominant Douglas-fir trees although most of the root biomass is
concentrated in the first 0.5 m (Shaw et al., 2014). The cumulative LAI is estimated to be
8-9 $m^2 m^{-2}$ (Parker et al., 2004). Additionally, the ecosystem hosts a large diversity of
mosses, lichens and other epiphytic plants, which play an important role in canopy OCS
dynamics (Rastogi et al., in revision). The soils are volcanic in origin, although most of
the forest surface is comprised of decaying organic matter (Shaw et al., 2004).
2.2. Study period: Measurements reported here are between April 18- Dec 31, 2015.
However, in early November an intake line at the top of the canopy was damaged after a
rainstorm. Measurements continued at the other intake heights (see sections 2.4 and 2.9).
Therefore ecosystem fluxes and related analyses in this study cover 136 days between
April 18 and October 31, while chamber based soil fluxes are reported for the months of
August-December. Gaps in the time series due to analyzer maintenance correspond to Jun
26-28, July 6-17, August 4-7, August 24 and October 4-7. April-October roughly
corresponds to most of the growing season, although at this site GPP usually peaks early
in March-April, when soil moisture is high and ecosystem respiration flux is low, while
plant productivity is typically severely light and temperature limited in the months of
November-December (Wharton and Falk, 2016). Environmental conditions during the
measurement campaign are shown in Figure 1 are represent a typical Mediterranean-type
climate, with temperature peaking in July and minimal to no measured rainfall between
June and September. This results in high summertime atmospheric vapor pressure deficit
(VPDa), and soil moisture declines steadily through the summer period, with some
recharge following rare precipitation events in September and then more commonly in
October. The measurement period also encompasses three distinct heat waves,
characterized by anomalously high air temperatures and mid-day VPDa values (often
exceeding 4 kPa). We examine the response of OCS and $CO_2$ fluxes during these heat
waves.





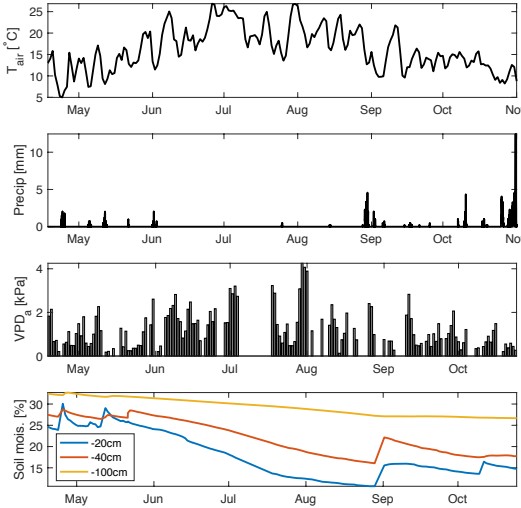

Figure 1. Environmental conditions at Wind River during the measurement campaign.
daily mean air temperature (a), precipitation (b) Mid-day VPDa (c) and Soil moisture
measured at three depths (d).
2.3. $CO_2$ and $H_2O$ eddy flux measurements: Carbon, water and energy fluxes have been
collected since 1998 at the Wind River AmeriFlux tower (US-wrc; Paw U et al. 2004).
For further details readers are referred to Falk et al., (2008; instrumentation and data
processing), and Wharton et al., (2012) and Wharton and Falk, (2016) for multi-year
carbon and water flux measurements and synthesis.
2.4. OCS measurements: A commercially available off-axis integrated cavity output
spectroscopy analyzer manufactured by Los Gatos Research Inc., (LGR; model 914-
0028) was deployed at the base of the tower in an insulated and temperature-controlled
shed. The instrument measures mixing ratios of OCS, $CO_2$, $H_2O$ and CO simultaneously
at a maximal scan rate of 5Hz. The system uses a 4.87 μm cascade laser coupled to a high
finesse 800 $cm^3$ optical cavity and light transmitted through the cavity is focused into a
cooled and amplified HgCdTe detector. OCS is detected at ~2050.40 $cm^{-1}$, $CO_2$ at
2050.56 $cm^{-1}$, CO at ~2050.86 $cm^{-1}$, and $H_2O$ at ~2050.66 $cm^{-1}$. Pressure broadening
associated with changes in the concentration of water vapor in the samples is corrected
for in the analysis routine. Air was sampled through 0.25'' diameter PFA tubing using a
diaphragm pump at a flow rate of 2L $min^{-1}$, from inlets located at 70m (at the height of
the eddy flux instrumentation), 60m (canopy top), 20m, 10m, and 1m. The sampling
frequency was 0.1Hz and the sampling interval was 5 minutes. The first minute of each
sampling interval was removed to avoid any inter-sampling mixing. The remaining data
were checked for temperature and pressure fluctuations inside the measurement chamber,
and a moving window filter was used to eliminate any sudden outliers in the data. Mixing
ratios were aggregated to provide hourly means. For detailed information regarding
instrumentation and the measurement readers are referred to Rastogi et al (in revision),
Berkelhammer et al. (2014) and Belviso et al. (2016).





2.5. Calibration: Calibration was performed using ambient air stored in insulated tanks as
a secondary reference. Air was sampled into the analyzer daily, and tank pressure was
routinely monitored to check for leaks. Glass flasks were randomly sampled form
calibration tanks and measured against a NOAA GMD reference standard. Cross-
referencing revealed that the accuracy of the measurement was within the reported
minimum uncertainty of the instrument (of 12.6 pmol mol$^{-1}$; Berkelhammer et al., 2016).
2.6. Thermal Camera measurements: Leaf temperatures were measured from October 28,
2014 to January 28, 2016 using a FLIR A325sc thermal camera (FLIR System Inc.,
Wilsonville, OR), in which a FLIR IR 30-mm lens (focal length: 30.38 mm; field of
view: 15°×11.25°) was installed. The thermal camera has a pixel resolution of 320 × 240.
Within the field of view (FOV), spot sizes of a single pixel are 0.83 cm from 10-m
distance and 8.3 cm from 100-m distance. Manufacturer-reported errors in original
measured thermal temperatures are ±2 °C or ±2% of the measurements. The camera
model is identical to one used in another study at an AmeriFlux site in central Oregon
(US Me-2), and the detailed specifications can be found in Kim et al. (2016). To monitor
a larger canopy region, a pan-tilt unit (PTU) was used for motion control, allowing
multiple canopy thermal image acquisition within one motion cycle. We used a FLIR
PTU-D100E (FLIR System Inc., Wilsonville, OR; (http://www.flir.com/mcs) to move the
thermal camera vertically and horizontally at specific pan and tilt angles. We selected
five pan-tilt angle (PT) positions representing the upper canopy (i.e., ~40 to 60 m above
the forest floor) to estimate leaf temperatures in this study.
2.7. Diffuse light measurement and analyses: An SPN1 Sunshine Pyranometer (Delta-T
Devices ltd., Cambridge, U.K.) was installed at the top of the canopy and collected direct
and diffuse shortwave downwelling radiation from April- December 2015. Measurements
were made every 1 min, and then aggregated to hourly means. We limited our analyses of
diffuse radiation data to include only mid-day hours (between 11am-1pm) to minimize
the influence of solar angles on diffuse radiation fractions. We defined three distinct
periods based on the ratio of diffuse radiation to total incoming solar radiation (*fidff*).
Data were characterized as clear if *fdiff* < 0.2; partly cloudy if *fdiff* > 0.2 and *fdiff* < 0.8,
and overcast if *fdiff* > 0.8.
2.8. OCS flux estimation: Canopy OCS flux was estimated using flux-gradient similarity,
following Commane et al., 2015.
$$F_{OCS} = F_{H2O} \cdot \frac{g_{ocs}}{g_{H2O}} \qquad (1)$$

where $F_{OCS}$, $F_{H2O}$, $g_{OCS}$ and $g_{H2O}$ are the fluxes and gradients of OCS and H$_2$O,
respectively. Following Seibt et al., (2010) and Berry et al., (2013), we assume that OCS
is irreversibly and rapidly consumed inside leaves, such that the gradient between
ambient air and the leaf interior effectively reduces to the ambient measured OCS mixing
ratio:
$$g_{ocs} = \chi_{ocs}^{a} - \chi_{ocs}^{l} = \chi_{ocs}^{a} \qquad , \qquad (3)$$

where $g_{OCS}$ is defined as he gradient of OCS between ambient air and the leaf
intercellular spaces ($\chi$ is the mixing ratio of OCS and superscripts *a* and *l* refer to ambient





and leaf respectively). In our study, $\chi^a_{OCS}$ is the measured mixing ratio at the canopy top
(60m) instead of above canopy (70m) to account for the boundary layer resistance, the
effect of which is likely low in tall and heterogeneous coniferous forests. We use vapor
pressure deficit (VPD) as the corresponding gradient for $H_2O$, under the key assumption
that the intercellular leaf surfaces are saturated with water vapor. While VPD is usually
calculated using air temperature, a more accurate calculation can be performed with leaf
temperatures, which can deviate significantly from air temperatures (Kim et al. 2016),
leading to significant differences between the VPD of ambient air and that at the leaf
surface (Fig. 2a and 3d in this study). Previously leaf temperatures have been inferred
from sensible heat fluxes, wind speed and air temperatures (e.g. Wehr et al., 2017), here
we use explicit measurements of leaf skin temperatures to estimate leaf-air VPD ($VPD_l$).
Analogous to Eq (3),
$$g_{H2O} = \chi^l_{H2O} - \chi^a_{H2O} = \frac{(e_s - e_a)}{P} = \frac{VPD_l}{P} \quad , \tag{4}$$
where $e_s$ is saturation vapor pressure at the leaf surface (kPa), using leaf skin temperature,
ea is the actual vapor pressure (kPa), P is the measured atmospheric pressure (Pa) at the
tower top, and $\chi^l_{H_2O}$ and $\chi^a_{H_2O}$ (ppth) are the leaf and ambient $H_2O$ mixing ratios at the
canopy top. Finally, since gradients of OCS and $H_2O$ are estimated between ambient air
and the leaf intercellular spaces, these are normalized by the ratio of diffusivities of these
two species in air (Seibt et al., 2010; Wohlfahrt et al., 2012).
$F_{H2O}$ was measured using eddy covariance at the tower top (70m). In high LAI forests
with minimal exposed soil, such as those of the Pacific Northwest, fluxes of $F_{H2O}$ can be
treated as a good proxy for transpiration, since soil evaporation should be minimal. We
excluded rainy days, as well as two days following rainfall, to only capture periods when
$F_{H2O}$ can be assumed to be dominated by transpiration. We included nighttime data since
several temperate tree species are known to transpire during the night (Dawson et al.,
2007). Moreover, in this particular forest OCS is taken up by epiphytes under conditions
of high humidity, which are common at nighttime (Rastogi et al., in revision). The first
term in right hand side of equation (1) was evaluated only under the condition $F_{H2O} > 0.2$
mmolm$^{-2}$s$^{-1}$. When this condition was not met (e.g. at nighttime), fluxes were calculated
using by integrating the rate of change in hourly OCS mixing ratios through the entire
profile.
Leaf Relative uptake was calculated following Seibt et al (2010).
$$LRU = \frac{F_{OCS}}{GPP} \cdot \frac{\chi CO_2}{\chi OCS} \quad , \tag{5}$$
where GPP was estimated from $CO_2$ fluxes measured at the tower top. Finally, canopy
conductance (Gc) was calculated by inverting the Penman Monteith equation (Monteith,
1965), which uses a combination of micrometeorological and eddy flux data collected
above the canopy at the tower top. Gc is the canopy-scale equivalent of stomatal
conductance, with the assumption that the canopy (or ecosystem) acts as a single big leaf.






$$Gc = \left[\frac{\rho Cp \delta e}{\gamma Le} + \frac{\left(\frac{\Delta}{\gamma}\right)\beta - 1}{Ga}\right]^{-1} \quad , \tag{6}$$

where ρ is air density (kg m$^{-3}$), Cp is specific heat (J kg$^{-1}$K$^{-1}$), δe is vapor pressure
deficit (kPa), Y is the psychrometric constant (kPa K$^{-1}$), Δ is the slope of the saturation
vapor pressure curve (kPa K$^{-1}$), β is the Bowen ratio (H:LE), and Ga is the aerodynamic
conductance for momentum transfer, calculated as u*$^2$.u$^{-1}$
(where u* is the friction velocity calculated from the momentum fluxes and u is the
horizontal wind speed). Ga provides a measure of how well the canopy top is coupled to
the background atmosphere (Wharton et al., 2012).
2.9. Surface Fluxes: A long-term automatic soil survey chamber (Li-Cor 8100-104, 20
cm diameter) was installed at three 0.03 m$^2$ surface sites in series, within 1 meter of each
other. All plastic and rubber parts had been removed from the chamber and replaced with
materials compatible with OCS measurements: stainless steel, PFA plastic, and Volara
foam. Blank measurements were performed in the laboratory before deployment and
OCS concentrations in the chamber were found to be indistinguishable from incoming
ambient concentrations. The stainless steel chamber top opened and closed automatically
on a timer. Gas was drawn through the chamber via a pump downstream of the analyzer,
and the 3 Lmin$^{-1}$ flow rate was confirmed with a mass flow meter. When the chamber
was open, ambient near-surface air was observed. When the chamber was closed, trace
gas concentrations reached a stable state for at least 2 minutes during the 10-minute
incubation time. The difference between the ambient concentration and the stable closed-
chamber concentration were used to calculate the surface fluxes of OCS and $CO_2$.
$$F_{forest\ floor} = M_c \Delta\chi . A^{-1} \quad , \tag{7}$$

where $M_c$ is the measured flow rate into the chamber (converted from Lmin$^{-1}$ to mols$^{-1}$
using the ideal gas law) and $\Delta\chi$ is the difference between mixing ratios of OCS or $CO_2$ in
ambient air and the chamber and A is the surface area of the chamber. The minimum flux
detectable with this method was 1.2 pmolm$^{-2}$s$^{-1}$ uptake or production.
Care was taken to select sites characteristic of the surface, which was generally springy
and covered in a mat of mosses and lichen. Surface flux observations were made at site 1
from July 6 to 16, site 2 from August 13 to October 7, and site 3 from November 6 to
December 2, 2015. The first site was visually similar to the subsequent two sites at the
surface, though the chamber base of the first site was installed into the moss layer and a
barely decomposed fallen tree. When a soil sample was attempted to be extracted from
the footprint of the chamber base, several liters of in tact wood litter were removed. The
influence of the developed soil on site 1 is therefore considered minimal. Site 2 was
selected nearby and observations were made until a dominant tree fell on the soil
chamber. The chamber was repaired and re-installed a month later at site 3 and
observations continued without incident until the chamber was removed in advance of the
soil freezing.
**3. Results and Discussion:**
3.1. Ecosystem fluxes: The composite diurnal cycles for $CO_2$, water vapor and OCS and
fluxes are shown (Fig. 2a-c). The total ecosystem flux of OCS ($F_{OCS}$; Fig 2.b.) follows a



pronounced diurnal cycle that peaks during daylight hours. The vertical profile of mixing
ratios measured throughout the canopy is also shown (right y-axis and orange lines in
Fig.2.b). OCS mixing ratios are highest at the canopy top and lowest near the forest floor,
but mixing ratios increase from the early morning to mid-afternoon. Together these
processes are indicative of ecosystem uptake and downward entrainment of boundary
layer air. While entrainment helps explain the diurnal cycle of observed mixing ratios,
this flux integrates to zero at daily and longer time scales (Rastogi et al., in revision). The
shape of the $F_{OCS}$ curve is very similar to those of net and gross carbon fluxes (Fig 2.b),
although $F_{OCS}$ was consistently negative throughout the 24-hour period, implying
ecosystem uptake during nighttime and daylight hours. While nighttime uptake of OCS
(mean nighttime flux ~ -10 ± 1 pmolm$^2$s$^{-1}$) is likely due to a combination of soil,
epiphyte, and vascular plant uptake due from partially closed stomata, daytime uptake is
likely dominated by vascular plant stomatal activity. Leaf relative uptake, a ratio of
$F_{OCS}$:GPP normalized by the mean mixing ratios of OCS:$CO_2$, showed a strong light
dependence. High-light, mid-day values ranged between 3-4, which is higher than those
observed at other forest systems (Kooijmans et al., 2017; Wehr et al., 2017) but well
within the spread of values obtained in a recent meta-analyses of OCS studies for
vegetated ecosystems (Whelan et al., 2018). The diurnal cycle was found to be
asymmetric, with peak values observed in the early morning, when stomatal conductance
is likely to be high (Winner et al., 2004), but GPP is limited by low light levels. It is
important to note that LRU is likely influenced by large amounts of epiphyte and
understory vegetation, which assimilate OCS even at times when ecosystem $CO_2$ uptake
is low or zero. Epiphytic assimilation of OCS is highly influenced by moisture content
(Gimeno et al., 2017) and is typically higher through the night and in the early mornings
at this site (Rastogi et al., in revision). Moreover, in tall old-growth forests, leaf area is
vertically distributed over a much larger part of the canopy compared to other forests
(Parker et al., 2004). While leaves at the canopy top exercise tight stomatal control to
limit water loss and minimize hydraulic failure (Woodruff et al., 2007) leaves lower
down in the canopy, including those of understory vegetation, likely impose less stomatal
control of transpiration (Winner et al., 2004). Lower-canopy leaves may therefore
continue to disproportionately assimilate OCS, even under low rates of carbon
assimilation (as $CO_2$ uptake is additionally light limited).





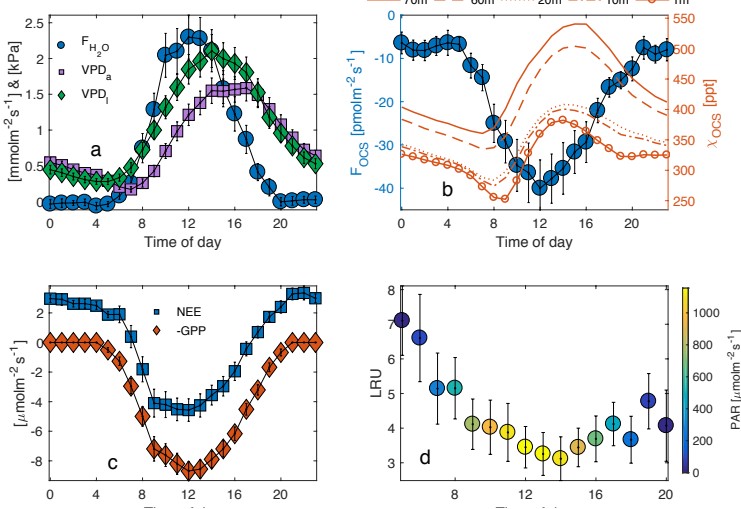

Figure 2. Diurnal cycle of $H_2O$ flux (blue circles) and VPD estimated from air and leaf
temperatures (purple squares and green diamonds respectively; a), estimated OCS flux
(circles, left axis) and mixing ratio profile (right axis; b), NEE and GPP (blue squares and
red diamonds; c), and leaf relative uptake (calculated only during daylight hours, colors
represent Photosynthetically active radiation; d).

3.2. Seasonal dynamics: Daytime fluxes of OCS and $CO_2$ followed similar patterns (Fig.
3a-b). Ecosystem uptake of OCS and $CO_2$ (as well as GPP) was highest in April, and
declined as the soil drought progressed. Mean monthly maximum OCS flux was
estimated as $-61 \pm 6$ pmol$^{-2}$s$^{-1}$, while daily mean maximum GPP over this period was
estimated as $10 \pm 1 \mu$molm$^{-2}$s$^{-1}$ (plotted as a negative quantity in Fig. 3b to show
ecosystem uptake). While the steepest declines in $F_{OCS}$, NEE and GPP happened between
the months of May and June, $F_{OCS}$ continued to decline through the rest of the summer,
with a minimum in August, incrementally increasing in September and decreasing again
in October. $CO_2$ fluxes flattened between June-September, before declining again in
October. During mid-late summer, water vapor flux declined (Fig.3c), as plants exercised
greater control over stomata responding to high VPD (peaking in July; Fig 3d). This can
be clearly seen in the seasonal cycle of canopy conductance (Gc; Fig. 3e) calculated
using the Penman-Monteith method. Mean monthly Gc was highest in the months of
April and May, and then declined precipitously with soil moisture, before increasing
again slightly in September and October following soil recharge and decreased VPD due
to precipitation events. At the monthly scale, patterns of daytime $F_{OCS}$ were most similar
to those of Gc and followed trends in soil moisture. In October, soil water recharge,
several rain-free days, and lower VPD (Fig. 1) do not result in increased gas exchange,
likely due to downregulation of photosynthesis (Eastman and Camm, 1995), induced by
photoprotective changes in the xanthophyll cycle (Adams and Demmig-Adams, 1994).





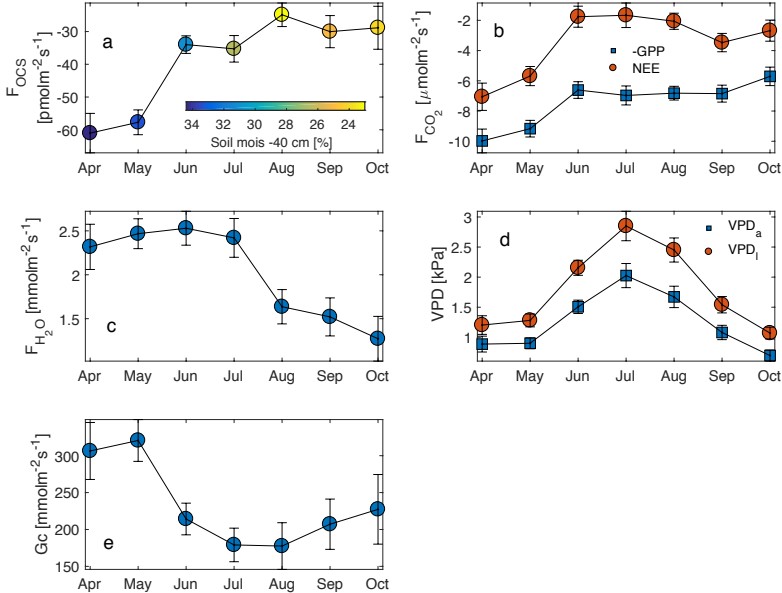


Figure 3. Monthly means for daytime $F_{OCS}$ colored according to soil moisture at 40cm
depth, NEE and -GPP (red circles and blue squares; b), water vapor flux (c), $VPD_a$ and
$VPD_l$ (blue squares and red circles; d), and canopy conductance (Gc; e).
3.3. Nighttime ecosystem and Surface Fluxes: While daytime fluxes of OCS and $CO_2$
were indicative of seasonal changes in ecosystem productivity and conductance, $F_{OCS}$ and
$F_{CO2}$ were driven by different environmental conditions during the night. Ecosystem
respiration is modeled based on temperature and therefore peaked in July (when air
temperature was highest). Nighttime $F_{OCS}$ however, was more related to soil moisture
status (blue circles in Fig. 4a-b). Nighttime $F_{OCS}$ was highest in April (mean = -12.7± 2.6
pmolm$^{-2}$s$^{-1}$), lowest between June and August (mean = -5.9± 1.5 pmolm$^{-2}$s$^{-1}$) and
increased again in October (mean = -9.7± 2.2 pmolm$^{-2}$s$^{-1}$). Nighttime uptake of OCS at
the site is likely due to soil (see below), epiphytes (Rastogi et al., in revision; Gimeno et
al., 2017), and incomplete stomatal closure (Kooijmans et al., 2017).
Forest floor OCS fluxes were observed from 3 sites in series and within 1 m of each
other. Site 1 had approximately twice the OCS uptake compared to the subsequent two
sites and had a substantial layer of intact woody debris under the chamber footprint. Site
2 and 3 had OCS fluxes similar to previous surface fluxes reported for forests (Whelan et
al., 2018). For all sites, there was no clear diurnal pattern. For site 2, uptake immediately
following chamber installation was higher (~6 pmol m$^{-2}$s$^{-1}$) than fluxes later on (all <6
pmol m$^{-2}$s$^{-1}$) when temperatures were lower. Site 3 did not have high uptake after
chamber installation, and had consistent fluxes between the detection limit and -6.2 pmol
m$^{-2}$s$^{-1}$ for the first few weeks. When ambient air temperatures dropped below freezing,
uptake remained unchanged, except for the largest uptake observed (6 to 12 pmol m$^{-2}$s$^{-1}$)
during two events when average air temperature fluctuated from a cooling to warming
trend. Soil temperature never dropped below freezing during the experiment and was





generally colder over time. We did not observe any OCS emissions from the chamber
based measurements, consistent with recent studies that find that cooler, moist (Maseyk
et al., 2014; Sun et al., 2016; Whelan et al., 2016) and radiation limited (Kitz et al., 2017)
soils do not emit OCS.
Surface $CO_2$ emissions exhibited a relationship with temperature, where highest
production (~25 µmol m$^{-2}$s$^{-1}$) corresponded with temperatures ~15°C, and maximum flux
values decreased for warmer and colder temperatures. $CO_2$ emissions had a diurnal
pattern, with lowest emissions at night and maximum emissions in late morning to mid
afternoon. No obvious relationship emerges from $CO_2$ emission and OCS uptake, though
the high OCS uptake events in late November and early December have a linear
relationship with $CO_2$ emissions. For sites 2 and 3, the ratio of OCS emission to $CO_2$
production, normalized by the concentration of OCS and $CO_2$ in the closed chamber, was
between -0.25 and -3.5 with a mean of -1. In contrast, the same ratio for site 1 varied
from -5 to -19 with a mean of -10.
At the peak of the soil drought (August; Fig. 1d), nighttime ecosystem OCS flux was
similar to the chamber-based surface fluxes, after which magnitudes differed by a factor
of 2-3. This difference can be explained by epiphytic consumption of OCS. Epiphytes are
a moisture dependent sink OCS at the site (Rastogi et al., in revision) and therefore are
likely inactive during the warmest and driest part of the year. Surface fluxes of $CO_2$ on
the other hand were much higher than ER estimated from the flux tower (blue circles in
Fig. 4b). While there are issues in scaling up chamber-based estimates, these results
corroborate earlier work that suggest that flux tower based estimates of ER at this site
might be underestimated (Harmon et al., 2004).

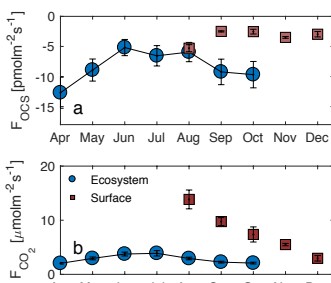

Figure 4. Nighttime ecosystem $F_{OCS}$ and $F_{CO2}$ (blue circles in a-b) and Surface $F_{OCS}$ and
$F_{CO2}$ from chamber measurements (brown squares in a-b) from sites 2 and 3. Site 1 was
atypical (see section 2.7) and therefore fluxes are not shown. Values for site 1 $F_{OCS}$ and
$F_{CO2}$ were -22 ± 0.3 pmolm$^{-2}$s$^{-1}$ and -83 ± 2 µmolm$^{-2}$s$^{-1}$ respectively.
3.4. Sensitivity to diffuse light: Mid-day fluxes of OCS and $CO_2$ were found to be
sensitive to changes in the fraction of diffused:total incoming shortwave radiation (*fdiff*;
Figure 5a-b). For these analyses, data were separated into three periods corresponding to
early summer (DOY 109-180), mid-late summer (DOY 180-240) and early fall (DOY
240-297), and binned into three categories: clear sky conditions, partly cloudy, and
overcast, defined in sec. 2.7. Mid-day VPD was highest under clear sky conditions and
lowest under overcast skies, but was most different across the three periods, during clear
skies (Fig. 5a). Consequently, OCS and $CO_2$ uptake was highest (most negative fluxes)





under overcast conditions during the early summer, and generally declined as *fdiff*
decreased across all time periods (Fig. 5b-d). Across the three periods, the rate of
decrease was much higher as *fdiff* changed from partially cloudy to clear. During the mid-
late summer, however, (red diamonds in Fig. 5a-f), the diffuse light effect resulted in
GPP and NEE being almost as high as during the early summer. $F_{OCS}$ was also highest
under partially cloudy skies during this time, and only showed a very weak decline under
completely overcast conditions. Overall, the behavior of OCS and $CO_2$ fluxes was similar
during the later time periods. Leaf relative uptake (LRU; calculated according to eq. 5)
was lowest under partly clear skies and highest under overcast conditions. This is because
under highly diffuse conditions, carbon uptake is additionally limited by light, whereas
$F_{OCS}$ is not (Wehr et al., 2017; Maseyk et al., 2014). The shape of the LRU curves can
additionally be explained by examining canopy conductance (Gc; Fig. 5f), which was
also higher under overcast skies. LRU increased with Gc across all three periods (Fig.
5g), and appeared to be constant for Gc greater than ~400 mmolm$^{-2}$s$^{-1}$.
The diffuse light enhancement of stomatal and canopy conductance is well documented
across a range of forest ecosystems (Alton et al., 2007; Cheng et al., 2015; Hollinger et
al., 2017; Urban et al., 2007; Wharton et al., 2012). Lower VPD (Fig. 5a) and light levels
allow plants to keep stomata open at mid-day and continue fixing $CO_2$. Lower VPD
reduces transpirational losses, and the lack of VPD-induced partial stomatal closure
reduces the resistance to $CO_2$ diffusion into the leaf. Correspondingly, the less directional
nature of diffuse solar radiation allows greater penetration into the canopy, thus
increasing photosynthesis across the entire canopy, even as a reduction in canopy top leaf
photosynthesis is observed due to a reduction in total radiation. In a multi-year analysis at
Wind River, Wharton et al., (2012) found that cloudy and partly cloudy sky conditions
during the peak-growing season lead to an enhancement of NEE. During our study, Gc
was generally higher in the early growing season, but increased as sky conditions
changed from clear skies to overcast. This increase was similar across the three time
periods, even as the response of OCS and $CO_2$ fluxes was different across these periods.
This indicates that declining soil moisture (Fig. 1d) likely limits gas exchange as the
summer progresses, even as canopy conductance can be reasonably high under overcast
skies.





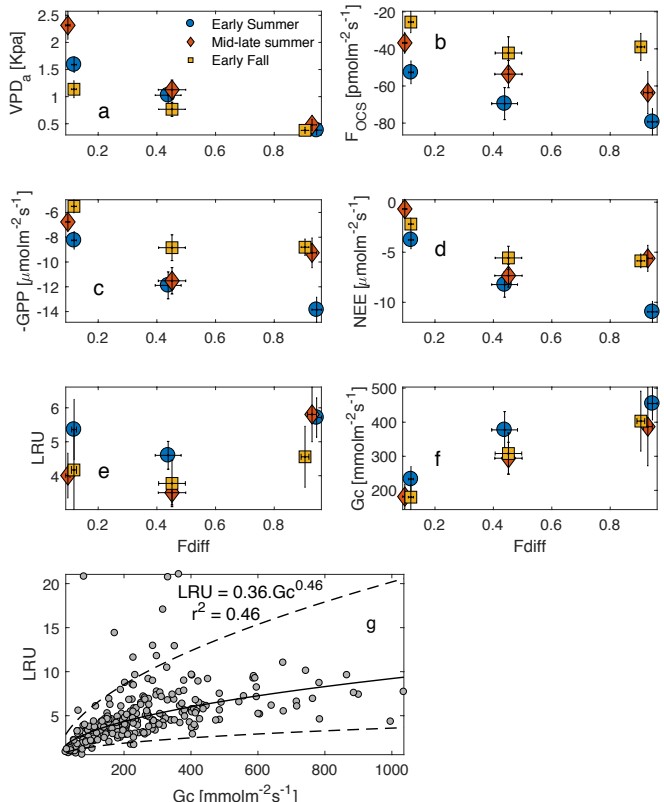


Figure 5. Mid-day VPDa, $F_{OCS}$, NEE and GPP plotted against the fraction of diffuse
downwelling shortwave radiation (a-d) for early summer, mid-late summer and early fall
of 2015 (these periods are defined in Section 3.4). High values on the x-axis indicate
completely overcast or cloudy conditions, whereas as low values indicated clear skies.
LRU increases with increasing *fdiff* during each period but the increase is most
pronounced in the early summer (e). Gc increases from clear to partly cloudy conditions
across the three periods and plateaus during overcast sky conditions (f). Across the three
periods, LRU increased with Gc, and levelled off at Gc values greater than ~ 400
mmolm$^{-2}$s$^{-1}$ (g).

3.5. Response to heat waves: 2015 was the warmest year in large parts of the Pacific
Northwest since records began in the 1930s (Dalton et al., 2017). We observed three
distinct heat waves during the 2015 summer. These were in early June (DOY 157-160),
end of June- early July (DOY 175-188) and late July-early August (DOY 210-213). The
three heat waves are shown as red, yellow and dark purple lines in Fig. 5; the overall time
series is shown in blue (mid-day means are plotted for all variables). Mid-day
temperatures exceeded 30°C during these heat wave events, while VPD-leaf exceeded
3Kpa during the first heat wave and increased to a maximum of 5.3 kPa during the last
event (Fig. 5b). During the first two events, $F_{OCS}$ was similar to days immediately prior
(Fig. 5c), but the canopy became a net source of $CO_2$ during all three events (Fig. 5d).





The third events lead to a severe reduction in $F_{OCS}$, even though the canopy had received
some rainfall in the preceding weeks (Fig. 1c). Water vapor fluxes (Fig 5e) increased
during the first two heat waves, compared to days immediately prior. The increased water
vapor is likely not from increased transpiration, as canopy scale stomatal conductance
during these events (Gc; Fig. 5f) is dramatically reduced. The increase is rather due to a
flux of water from the soil surface and epiphytes that can store water in the canopy. High
temperatures during such events are likely to result in increased evaporation.

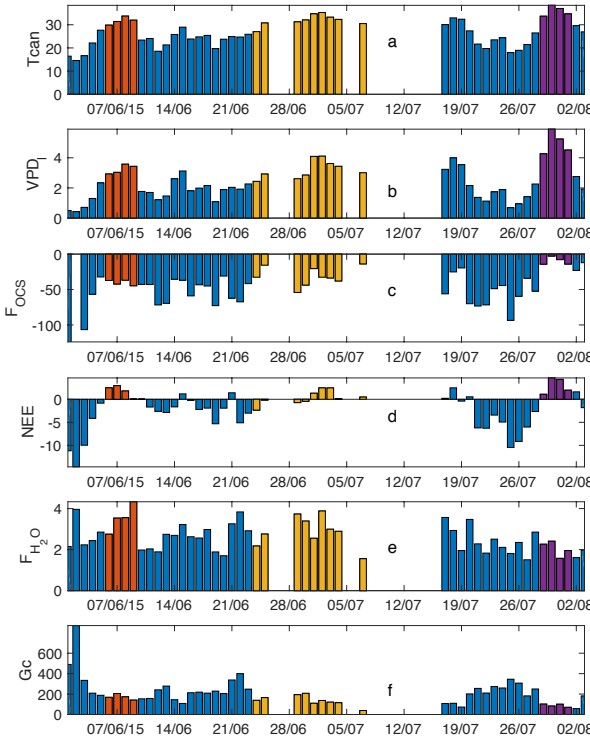


Figure 6. Mid-day means (11am-1pm local time) for three heat wave periods (plotted as
red, yellow and purple, while the overall time series is shown in blue). Variables
displayed are canopy temperature (°C; a), VPD-leaf (b), $F_{OCS}$ (c), NEE (d), water vapor
flux (e), and canopy conductance (Gc, f). Units for each panel are the same as specified
in previous figures.
**4. Conclusions**
Over hourly and seasonal timescales, estimates of $F_{OCS}$ generally tracked fluctuations in
GPP, implying stomatal control of carbon, water, and OCS fluxes at the site. We used
continuous in-situ measurements of OCS mixing ratios, collocated measurements of
water vapor fluxes, and air and canopy temperatures to calculate OCS uptake. We found
the forest to be a large sink for OCS, with sink strength peaking during daylight hours.
The mean LRU was ~ 4, and varied in response to changing light conditions and canopy



conductance. These LRUs are larger than observed from other ecosystem scale studies,
but well within the range of reported values (Whelan et al., 2018; Sandoval-Soto et al.,
2005). The forest surface was found to be a soil moisture dependent sink of OCS, with
magnitudes that were roughly half of nighttime ecosystem fluxes, indicating other
components of the ecosystem (epiphytes present throughout the canopy and impartial
stomatal closure) to also take up OCS during these hours.  Ecosystem fluxes of OCS and
$CO_2$ were found to be strongly sensitive to the ratio of diffuse:direct radiation reaching
the top of the canopy. Uptake of both OCS and $CO_2$ increased as sky conditions changed
from clear to partly cloudy. A much smaller increase in uptake was observed as sky
conditions changed from partly cloudy to overcast, except during the early summer, when
soil moisture was not limiting. This change was mediated by the sensitivity of stomata to
changing cloudiness and soil moisture, as estimated from canopy conductance calculated
according to the inverted Penman-Monteith equation. Finally we examined the response
of OCS, $CO_2$ and $H_2O$ fluxes on heatwaves, and found that sequential heatwaves lead to
suppression in stomatal gas exchange of all three fluxes.
Our results support the growing body of work that suggests ecosystem-scale OCS uptake
is controlled by stomatal dynamics. While moist old-growth forests in Pacific
Northwestern U.S. do not represent a very large fraction of the global terrestrial surface
area, results from this study are likely relevant for other old-growth forests, particularly
high LAI and very wet forests with extensive epiphyte cover, which are widespread in the
humid tropics.
Acknowledgements:
This work was partly funded by NASA SBIR Phase II award NNX12CD21P to LGR,
Inc. ("Ultrasensitive Analyzer for Realtime, In-Situ Airborne and Terrestrial
Measurements of OCS, CO2, and CO."). We would like to thank the US Forest Service
and the University of Washington for letting us use the research facility at Wind River. In
particular, we wish to sincerely acknowledge Ken Bible and Matt Schroeder for their help
with setting up the experiment as well as maintenance throughout the measurement
campaign. Data collected and used in this study can be accessed at
ftp.fsl.orst.edu/rastogib/Biogeosciences2018_Rastogi.

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
