# Peer review of "Biogeosciences Discuss., https://doi.org/10.5194/bg-2018-85 Manuscript under review for journal Biogeosciences"

_Biogeosciences, 2018_

## Referee Comment (RC1) · Anonymous Referee #1 · 2 May 2018

The manuscript reports a very interesting and important research concerning the relationship between carbonyl sulfide and carbon dioxide fluxes on ecosystem level and their response to diffuse radiation and heat waves. Current literature lacks the flux measurements provided by this study. This study will help fill in some knowledge gaps needed for implementing carbonyl sulfide as a constrain for the gross primary production on larger scales. Although the author had good references concerning the methods it lacks some basic information. Is the used instrument capable of correcting for the any effects due to water broadening and equally important were the surfaces chamber fluxes executed correctly? If the flow that is sucking the air out of the chambers is too high, COS depleted air from lower soil layers could distort the measurements.

[Figure]

Concerning the concentration gradient, I was wondering why no invers lagrangian modelling was done as this method could help determine the sinks or sources within the canopy.

30-33 This statement is a bit farfetched. On what basis do you make this statement? LRU varies in your study, not only between seasons, but also as a result of changing light conditions (fraction of diffuse downwelling shortwave radiation)

48 This is not entirely true, under stressed conditions plants have been reported to emit COS. Add:

Bloem, E., et al. (2012). "Sulfur Fertilization and Fungal Infections Affect the Exchange of H2S and COS from Agricultural Crops." Journal of Agricultural and Food Chemistry 60(31): 7588-7596. or other stress related OCS publication, as a heatwave might change the ratio of OCS to $CO_2$ uptake.

115 If related to plant stress and photosynthesis (108), water potential would be a much better parameter to reflect the plant available water (if the parameter is available). Plant available water strongly depends on soil type and structure.

140 My knowledge about the Los Gatos instrument is limited, but as literature tells me, the build in water correction of the instrument might not able to fully compensate for the effect of water vapor in sample air. Have you done dependency curves of gas with a known OCS concentration at levels of different water vapor to test your instrument and the analysis routine? If not, I would strongly suggest doing this to avoid or correct for measurement errors. For further information, I recommend reading: Bunk, R., et al. (2017). "Exchange of carbonyl sulfide (OCS) between soils and atmosphere under various $CO_2$ concentrations." Journal of Geophysical Research-Biogeosciences 122(6): 1343-1358. See section 2.3, where this problem has been tackled with!

176 A reference suggesting only using mid-day hours would be appreciated. Didn't the cloud cover change from early morning to late evening?

[Figure]

181 Did you have problems applying the modified bowen ratio method? The publication cited in Commane et al. 2015 Meyers, T. P., et al. (1996). "Use of the modified Bowen-ratio technique to measure fluxes of trace gases." Atmospheric Environment 30(19): 3321-3329 States, that using this method might have issues when used within plant canopies. They state that "Infrequent but large energetic eddies are responsible for most of the exchange that occurs within canopies (Baldocchi and Meyers, 1991; Shaw et al., 1983). Transport by these coherent structures often leads to the counter-gradient flux structure frequently observed in crop and forest canopies." Also, why didn't you apply invers Lagrangian modelling like: Nemitz, E., et al. (2000). "Sources and sinks of ammonia within an oilseed rape canopy." Agricultural and Forest Meteorology 105(4): 385-404.

and

Karl, T., et al. (2004). "Exchange processes of volatile organic compounds above a tropical rain forest: Implications for modeling tropospheric chemistry above dense vegetation." Journal of Geophysical Research: Atmospheres (1984–2012) 109(D18).

You could even get the information about the source or sink strength of layers within your canopy.

232 Even though you reference Falk et al. (2008) state that you are using a night time flux partitioning method that has been optimized to the field site. The LRU in this study will be used by modelers and I think the information from what the LRU is calculated is crucial.

246 Are you using the Licor 8100 as flow through chamber with ambient air able to enter the chamber while you suck out the air at another end? If so, is the flowrate of 3 liters per minute not too much? How big were the openings of the chamber where ambient air was allowed to enter the chamber? If the flowrate is too high, air would be sucked out of the soil which would alter the fluxes you measure. Have you done differential pressure measurements like: Kitz, F., et al. (2017). "In situ soil COS exchange of a

temperate mountain grassland under simulated drought." Oecologia: 1-10.

251 Was it statistically indistinguishable, then write so.

286 Fig 2b-c instead of 2b

307 Invers lagrangian modelling could answer this question. Again, why not apply it?

311 Fig 2c add FCO2 to ylabel

346 There is no soil moisture in plot 4 which would help the reader see this correlation. Is there statistical evidence or just a trend?

363 To make this statement you would have to compare the soil fluxes of your site with the publications. In your case, you have a combination of soil plus understory plants and mosses which could compensate for a soil emission. (As you stated in line 271: "The influence of the developed soil on site 1 is therefore considered minimal."). I would use the citation you used to tell that no soil emissions are expected at your site.

373 In line 363 you write that you haven't observed any OCS emission, I guess you meant uptake in line 373?

388 Please state what the error bars stand for (I assume standard deviation).

394 5b-c (a would be VPD)

424 As NEE includes both, GPP and RECO, are you saying both components are increasing during the peak growing season, or did you want to refer to the CO2 uptake only?

123 mid-day VPDa (c) and soil moisture instead of Mid-day VPDa (c) and Soil moisture

225 When this condition was not met (e.g. at nighttime), fluxes were calculated by integrating the rate of change in hourly OCS mixing ratios through the entire profile. —skip using

152 from instead of form

---

## Referee Comment (RC2) · Anonymous Referee #2 · 5 May 2018

General Comments

Rastogi et al. present observed patterns of OCS uptake in an old-growth forest during 2015. Their observations are consistent with previous studies in similar ecosystems, and are valuable in corroborating those studies and in confirming that the community's general understanding of OCS uptake holds in old-growth forests. The methods seem valid, subject to some concerns noted below. The manuscript is easy to read and clearly organized. This is not a manuscript that presents new insights or methods.

Unfortunately, the inferences drawn from the observations in the manuscript are not quantitatively supported. In particular, the inferences about stomatal responses to soil

moisture and heat waves seem to be not only unsupported but also incorrect (see below).

The core weakness of the manuscript, which contributes to the inference problem just mentioned, is that it is overly descriptive in terms of both the analysis and the writing. In terms of the analysis, 5 of the 6 figures (and all but subsection 3.4 of the text) present time series of data, and the associated analysis is restricted entirely to 'eyeballing' correlations between those time series. The authors do not calculate correlation coefficients, or use multiple regression or a simple model to support their causal inferences. In terms of the writing, many patterns in the data are described in the text even though they aren't clearly connected to any conclusions. The manuscript would be more effective if it were to focus on what was learned from the data, referring to the data as necessary to support those findings. Other patterns could be gleaned from figures or tables by any reader with a particular interest.

Specific Comments

- lines 193-195: This justification doesn't make sense to me. The resistance to turbulent eddy transport through open air from 70m to 60m should be much less than the resistance to eddy transport through the dense canopy from 60m to the leaf surfaces. If the aim is to establish the gradient across only the stomata, then using 60m instead of 70m hardly helps. The full transport resistance from the tower top to the substomatal cavity of some particular leaf is $r\_ac + r\_wc + r\_lbl + r\_s$, where $r\_ac$ and $r\_wc$ are the above-canopy and within-canopy turbulent eddy resistances, $r\_lbl$ is the leaf boundary layer resistance, and $r\_s$ is the stomatal resistance. Of these, $r\_ac$ is probably negligible, $r\_s$ is probably most limiting, and $r\_lbl$ is probably second most limiting. The authors appear to have neglected $r\_lbl$ and $r\_wc$, so that their Gc is not exactly the canopy-scale stomatal conductance but rather a canopy-scale combination of the stomatal, leaf boundary layer, and within-canopy turbulent eddy conductances, i.e. $Gc = 1/(r\_wc + r\_lbl + r\_s)$. It is possible to measure $r\_ac$ and some portion of $r\_wc$ by comparing concentration measurements within the canopy to those above the

canopy and using flux = conductance x gradient; this approach ought to be superior to using the theoretical u*^2/u. In any case, the authors should clarify what they mean by "the" boundary layer, as the boundary layer that is usually discussed in the context of stomatal uptake is the leaf boundary layer, i.e. the thin layer of stagnant air against the surface of individual leaves, through which gas transport is substantially diffusive rather than advective or convective. Transport through the canopy airspace, or the 10m above the canopy, on the other hand, will not be diffusive at all.

- line 223: You are talking about computing the change in canopy storage, which is a good idea, but why not do that at all times? In general, the flux through the stomata should be equal to the flux past the eddy flux sensor plus the flux into the canopy storage airspace, i.e. Eq. 1 should have a storage term appended to the right hand side (perhaps you used to have one there, as suggested by your reference to the "first term in right hand side of equation (1)" on line 221?). Here you are saying that when the eddy flux term was near zero, you considered the storage flux term. But the storage flux term might be substantial even when the eddy flux term is not near zero.

- line 236 (Eq. 6): Given the considerations about energy imbalance and the PM equation raised by Wohlfahrt et al., 2009 (Agricultural and Forest Meteorology 149, 1556–1559) and by Wehr et al., 2017 (Biogeosciences 14, 389–401), it should be stated why this particular form of the PM equation was used (or why the PM equation was used at all instead of just using sensible heat flux measured from the tower). Those papers indicate that retrieved values of stomatal conductance can be substantially affected by the choice of equation.

- line 329: Regarding "declined precipitously with soil moisture", it is a bit hard to tell from the color scale in Fig. 3a, but it looks like the decline in OCS (which matches the decline in Gc), is better correlated with the rise in VPD than with the drop in soil moisture. People often assume that soil moisture drives seasonal patterns in stomatal conductance (and it surely does at some sites), but it is also possible that the seasonal pattern in Gc and F_OCS is explained entirely by VPD (that was the finding for the

mesic Harvard Forest site used in the Wehr et al., 2017 paper you cite). It would be interesting to try to disentangle those two water-related drivers here, at least with a simple regression approach.

- line 380-1: Did the estimation of ER from the tower include measurements of canopy $CO_2$ storage? If turbulence is low at night, most of the respired $CO_2$ is probably accumulating in airspaces below the eddy flux sensor.

- line 427-9: I don't see how this inference is connected to the preceding observations, and I don't see any evidence in the manuscript that soil moisture (as opposed to VPD) is limiting gas exchange.

- line 452-5 and 483-4: These inferences are flawed. It is not true that "canopy scale stomatal conductance during these events is dramatically reduced". Figure 6 shows that Gc was not reduced at all during the first heat wave, and was not reduced until the end of the second heat wave, at which point the water flux also dropped. During the third heat wave, Gc was reduced, but the water flux did not increase. Even more importantly, Gc was estimated based on the assumption that the water flux was exclusively transpiration, so it makes no sense to say that the behavior of Gc implies the increased water flux was not transpiration. If the approach used to calculate Gc is valid, then the increased water flux was indeed due to increased transpiration, on account of an increased VPDL.

Technical Corrections

- line 23: "established theoretical relationships" is vague and perhaps overly confident; better to say "we employ the flux-gradient method to infer" - line 109: should be "severely light- and temperature-limited" - line 45: add a "the" before "branch scale" or remove the "the" before "ecosystem scale" for grammatical parallelism - line 111: should be "are shown in Figure 1 and" - line 122: fix capitalizations (should be "Daily mean air temperature (a), precipitation (b) mid-day VPDa (c) and soil moisture" - line 127: citation format is flawed (don't put comma and parentheses) - line 236 (Eq. 6):

variables need cleaning up (e.g. Cp should have p as subscript and same for delta_e), and the description of variables below the equation needs to match the equation (e.g. Y is not gamma). - line 221: should be "the water flux in equation (1) was" (the right side of Eq. 1 has only one term, containing three variables) - line 269: should be "intact" - line 276: delete "and" at end of line - line 342-3: nighttime ecosystem was measured by EC, not modeled, if I understand correctly - line 352 "Site" should be plural at end of line - line 392: "diffused" should be "diffuse" - line 397: remove comma between "periods" and "during" - line 398: "was" should be "were" - line 392: again, variables have to stay consistent in capitalization, italicization, subscript, etc. (fdiff). - line 418: "Correspondingly" doesn't fit. Should be "Additionally". - line 444: should be Fig. 6, not Fig. 5 (and throughout this paragraph) - line 447: "3Kpa" should be "3 kPa" - line 450: "the third events lead" should be "the third event led"

---

## Author Comment (AC1) · 2 Aug 2018

**Response to Reviewer 1**

The manuscript reports a very interesting and important research concerning the relationship between carbonyl sulfide and carbon dioxide fluxes on ecosystem level and their response to diffuse radiation and heat waves. Current literature lacks the flux measurements provided by this study. This study will help fill in some knowledge gaps needed for implementing carbonyl sulfide as a constrain for the gross primary production on larger scales. Although the author had good references concerning the methods it lacks some basic information. Is the used instrument capable of correcting for the any effects due to water broadening and equally important were the surfaces chamber fluxes executed correctly? If the flow that is sucking the air out of the chambers is too high, COS depleted air from lower soil layers could distort the measurements. Concerning the concentration gradient, I was wondering why no invers lagrangian modelling was done as this method could help determine the sinks or sources within the canopy.

We thank the reviewer for providing thoughtful and detailed feedback on the manuscript. We address comments about water broadening, surface measurements and Lagrangian modelling in this response.

30-33 This statement is a bit farfetched. On what basis do you make this statement? LRU varies in your study, not only between seasons, but also as a result of changing light conditions (fraction of diffuse downwelling shortwave radiation)

Thank you for pointing this out. We have changed these lines in the abstract (lines-25-29) to say "*OCS fluxes showed a pronounced diurnal cycle, with maximum uptake during mid-day. OCS uptake was found to scale with independent measurements of $CO_2$ fluxes over a 60-m-tall old-growth forest in the Pacific Northwestern U.S. (45°49'13.76" N; 121°57'06.88") at daily and monthly timescales under mid-high light conditions across the growing season in 2015.*"

48 This is not entirely true, under stressed conditions plants have been reported to emit COS. Add:

Bloem, E., et al. (2012). "Sulfur Fertilization and Fungal Infections Affect the Exchange of H2S and COS from Agricultural Crops." Journal of Agricultural and Food Chemistry 60(31): 7588-7596. or other stress related OCS publication, as a heatwave might change the ratio of OCS to CO2 uptake.

Thank you for providing the reference. We have included this (line 49).

115 If related to plant stress and photosynthesis (108), water potential would be a much better parameter to reflect the plant available water (if the parameter is available). Plant available water strongly depends on soil type and structure.

We agree with the reviewer's comment about water potential. Unfortunately, we were not able to measure this quantity at the time of measurement.

140 My knowledge about the Los Gatos instrument is limited, but as literature tells me, the build in water correction of the instrument might not able to fully compensate for the effect of water vapor in sample air. Have you done dependency curves of gas with a known OCS concentration

at levels of different water vapor to test your instrument and the analysis routine? If not, I would strongly suggest doing this to avoid or correct for measurement errors. For further information, I recommend reading: Bunk, R., et al. (2017). "Exchange of carbonyl sulfide (OCS) between soils and atmosphere under various CO2 concentrations." Journal of Geophysical Research-Biogeosciences 122(6): 1343-1358. See section 2.3, where this problem has been tackled with!

Thank you for pointing this. This was a cause of concern, we worked with the manufacturer on this, and found that while there was a slight dependence of water vapour on OCS mixing ratios, the effect was very small and likely doesn't affect measurements in this study (Fig.1).

[Figure]

Figure 1. OCS water vapor cross-interference tested in the lab. The magnitude of this cross interference is negligible compared to observed canopy- atmosphere ecosystem exchange in this study

176 A reference suggesting only using mid-day hours would be appreciated. Didn't the cloud cover change from early morning to late evening?

Cloud cover does change during the day. However, the ratio of direct: diffuse light is also sensitive to solar zenith angles (such that the fraction of diffused light is always higher in the mornings and evenings). This is why we restrict out analyses with diffused radiation to mid-day.

181 Did you have problems applying the modified bowen ratio method? The publication cited in Commane et al. 2015 Meyers, T. P., et al. (1996). "Use of the modified Bowen- ratio technique to measure fluxes of trace gases." Atmospheric Environment 30(19): 3321-3329 States, that using this method might have issues when used within plant canopies. They state that "Infrequent but large energetic eddies are responsible for most of the exchange that occurs within canopies (Baldocchi and Meyers, 1991; Shaw et al., 1983). Transport by these coherent structures often leads to the counter-gradient flux structure frequently observed in crop and forest canopies." Also, why didn't you apply invers Lagrangian modelling like: Nemitz, E., et al. (2000). "Sources and sinks of ammonia within an oilseed rape canopy." Agricultural and Forest Meteorology 105(4): 385-404. and

Karl, T., et al. (2004). "Exchange processes of volatile organic compounds above a tropical rain

forest: Implications for modeling tropospheric chemistry above dense vegetation." Journal of Geophysical Research: Atmospheres (1984–2012) 109(D18).

You could even get the information about the source or sink strength of layers within your canopy.

Thank you for your suggestions and accompanying references. We have been trying to use an inverse Lagrangian/ Eulerian model to address this very problem. This is an ongoing project, we have had severe challenges in parametrizing a model that is able to estimate the wind profile and other parameters (such as eddy diffusivity) through the tall old-growth canopy. We hope to address this, and report these in a future study.

232 Even though you reference Falk et al. (2008) state that you are using a night time flux partitioning method that has been optimized to the field site. The LRU in this study will be used by modelers and I think the information from what the LRU is calculated is crucial.

We agree and have included this information (lines 228-229).

246 Are you using the Licor 8100 as flow through chamber with ambient air able to enter the chamber while you suck out the air at another end? If so, is the flowrate of 3 liters per minute not too much? How big were the openings of the chamber where ambient air was allowed to enter the chamber? If the flowrate is too high, air would be sucked out of the soil which would alter the fluxes you measure. Have you done differential pressure measurements like: Kitz, F., et al. (2017). "In situ soil COS exchange of a temperate mountain grassland under simulated drought." Oecologia: 1-10.

Yes, we agree with the reviewer on the issue of high flow rate. However, in contrast to other chambers, there is a vent on the top of the Li-Cor 8100 soil chamber that allows equalization of pressure between the inside and outside of the chamber. This consists of an always-open tube with a specially designed flow path to keep pressures stable even in windy conditions (Xu et al., 2006).

Xu, L., Furtaw, M. D., Madsen, R. A., Garcia, R. L., Anderson, D. J., & McDermitt, D. K. (2006). On maintaining pressure equilibrium between a soil $CO_2$ flux chamber and the ambient air. *Journal of Geophysical Research Atmospheres*, *111*(8), 1–14. https://doi.org/10.1029/2005JD006435

251 Was it statistically indistinguishable, then write so.

Yes, thank you. We have included this (line 241).

286 Fig 2b-c instead of 2b

Changed accordingly.

307 Inverse lagrangian modelling could answer this question. Again, why not apply it?

Kindly see comment above.

311 Fig 2c add FCO2 to ylabel

Changed accordingly.

346 There is no soil moisture in plot 4 which would help the reader see this correlation. Is there statistical evidence or just a trend?

We have included a figure (Fig. 3) that shows this relationship.

363 To make this statement you would have to compare the soil fluxes of your site with the publications. In your case, you have a combination of soil plus understory plants and mosses which could compensate for a soil emission. (As you stated in line 271: "The influence of the developed soil on site 1 is therefore considered minimal."). I would use the citation you used to tell that no soil emissions are expected at your site.

We have included the relevant citation (lines 354-355).

373 In line 363 you write that you haven't observed any OCS emission, I guess you meant uptake in line 373?

We mean to say that OCS uptake was correlated with $CO_2$ emissions.

388 Please state what the error bars stand for (I assume standard deviation).

Noted accordingly (line 371).

394 5b-c (a would be VPD)

Changed accordingly.

424 As NEE includes both, GPP and RECO, are you saying both components are increasing during the peak growing season, or did you want to refer to the CO2 uptake only?

We have changed this to "increase in CO2 uptake" (line 406).

123 mid-day VPDa (c) and soil moisture instead of Mid-day VPDa (c) and Soil moisture

Changed accordingly.

225 When this condition was not met (e.g. at nighttime), fluxes were calculated by integrating the rate of change in hourly OCS mixing ratios through the entire profile. —skip using

We have removed storage flux estimates from estimation of canopy-scale leaf OCS flux.

152 from instead of form

Changed accordingly.

---

## Author Comment (AC2) · 2 Aug 2018

Response to reviewer 2

"Rastogi et al. present observed patterns of OCS uptake in an old-growth forest during 2015. Their observations are consistent with previous studies in similar ecosystems, and are valuable in corroborating those studies and in confirming that the community's general understanding of OCS uptake holds in old-growth forests. The methods seem valid, subject to some concerns noted below. The manuscript is easy to read and clearly organized. This is not a manuscript that presents new insights or methods."

We thank the reviewer for a very thoughtful and detailed response to our submitted manuscript. However, we would like to disagree that this method presents no new insights or methods. In this work, we propose a simple model to estimate ecosystem- scale leaf OCS fluxes from concentration measurements, using other novel measurements, such as direct measurements of canopy skin temperature, using a thermal camera. This model is based on a theoretical framework laid out by Seibt et al., (2010) and Wohlfahrt et al., 2012), and supported by other seminal ecosystem-scale studies relating OCS uptake to plant productivity or GPP (Commane et al., 2015) and stomatal conductance (Wehr et al., 2017). In addition, we show the response of inferred OCS fluxes to the diffuse fraction of downwelling radiation, as well as the response of OCS fluxes to sequential heatwaves. These responses have not been reported for any ecosystems as yet, and we hope they provide important constraints on the use of OCS as a tracer for stomatal conductance and ultimately GPP.

"Unfortunately, the inferences drawn from the observations in the manuscript are not quantitatively supported. In particular, the inferences about stomatal responses to soil moisture and heat waves seem to be not only unsupported but also incorrect (see below). The core weakness of the manuscript, which contributes to the inference problem just mentioned, is that it is overly descriptive in terms of both the analysis and the writing. In terms of the analysis, 5 of the 6 figures (and all but subsection 3.4 of the text) present time series of data, and the associated analysis is restricted entirely to 'eyeballing' correlations between those time series. The authors do not calculate correlation coefficients, or use multiple regression or a simple model to support their causal inferences. In terms of the writing, many patterns in the data are described in the text even though they aren't clearly connected to any conclusions. The manuscript would be more effective if it were to focus on what was learned from the data, referring to the data as necessary to support those findings. Other patterns could be gleaned from figures or tables by any reader with a particular interest."

We appreciate these suggestions and have reworked specific parts of the manuscript to provide more quantitative comparisons, as well as changed the language of our study that relates to soil moisture. We have also tidied up the manuscript so that it reads more cleanly.

Specific Comments:

"- lines 193-195: This justification doesn't make sense to me. The resistance to tur- bulent eddy transport through open air from 70m to 60m should be much less than the resistance to eddy transport through the dense canopy from 60m to the leaf sur- faces. If the aim is to establish the gradient across only the stomata, then using 60m instead of 70m hardly helps. The full transport resistance from the tower top to the substomatal cavity of some particular leaf is r_ac + r_wc + r_lbl + r_s, where r_ac and r_wc are the above-canopy and within-canopy turbulent eddy resistances, r_lbl is the leaf boundary layer resistance, and r_s is the stomatal resistance. Of these, r_ac is probably negligible, r_s is probably most limiting, and r_lbl is probably second most limiting. The authors appear to have neglected r_lbl and r_wc, so that their Gc is not exactly the canopy-scale stomatal conductance but rather a canopy-scale combination of the stomatal, leaf boundary layer, and within-canopy turbulent eddy conductances, i.e. Gc = 1/(r_wc + r_lbl + r_s). It is possible to measure r_ac and some portion of r_wc by comparing concentration measurements within the canopy to those above the canopy and using flux = conductance x gradient; this approach ought to be superior to using the theoretical u*^2/u. In any case, the authors should clarify what they mean by "the" boundary layer, as the boundary layer that is usually discussed in the context of stomatal uptake is the leaf boundary layer, i.e. the thin layer of stagnant air against the surface of individual leaves, through which gas transport is substantially diffusive rather than advective or convective. Transport through the canopy airspace, or the 10m above the canopy, on the other hand, will not be diffusive at all. "

We would again like to thank the reviewer for such a carefully detailed and clear comment regarding conductance. We have rephrased this text in the original manuscript, which we acknowledge was incorrect. We agree with the reviewer that the transport between 70-60m is in fact turbulent (and therefore more related to r_ac than to r_lbl). The choice to use the canopy top mixing ratios is following previously published literature (Fares et al., 2012; and references therein). We are, however, not ignoring the leaf boundary layer in our formulation (i.e. eqs, 1-3). Here we argue (following previous studies, cited above) that the ratio of fluxes of two scalars (in this case, OCS and $H_2O$) across the leaf surface is proportional to the gradient between the ambient air and the leaf intercellular spaces, i.e.

$$\frac{F_{OCS}}{F_{H2O}} = \frac{OCS_a - OCS_i}{(e_i - e_a) \cdot P^{-1}} \cdot 1.94^{-1} \tag{1}$$

where $F_{OCS}$ and $F_{H2O}$ are fluxes of OCS and $H_2O$ (in units of $pmol\,m^{-2}s^{-1}$ and $mmol\,m^{-2}s^{-1}$ respectively), $OCS_a$ and $OCS_i$ represent ambient and intercellular mixing ratios of OCS respectively (ppt), where $VPD_l = e_i - e_a$ ($e_i$ and $e_a$ are intercellular and actual vapor pressure; kPa), and P is atmospheric pressure (kPa). We had incorrectly labelled saturated leaf pressure

obtained from leaf temperatures as $e_s$, and have now correctly labelled this as $e_i$ (since it is the leaf intercellular spaces that are assumed to be saturated with water vapor, not the leaf surface). Finally, the factor 1.94 reflects the diffusivity ratio of OCS and $H_2O$.

To investigate the reviewer's concern regarding various resistances, we additionally estimated OCS fluxes according to the following equation

$$F_{OCS} = \left(\frac{1.56}{G_{bw}} + \frac{1.94}{G_{sw}}\right)^{-1} . OCS_a \tag{2}$$

where $G_{bw}$ and $G_{sw}$ are the canopy- scale boundary layer and stomatal conductances for water vapour transport. The constants 1.56 and 1.94 are the ratios of diffusivities of OCS and $H_2O$ under turbulent and diffusive flow (Seibt et al., 2010).

We derived $G_{bw}$ by first estimating roughness parameters following Monin-Obukhiv similarity theory (Foken, 2006). These were then used to obtain stability parameters for momentum transport, which was finally used to estimate $G_{bw}$ following Su et al., (2001). Code and further information can be found within the R package "bigleaf" and accompanying manual (https://bitbucket.org/juergenknauer/bigleaf). $G_{sw}$ was estimated as

$$G_{sw} = (G_{cw}^{-1} - G_{bw}^{-1})^{-1} \tag{3}$$

where $G_{cw}$ is the canopy (surface) conductance to water vapor transport. To address the reviewer's comments regarding the use of the Penman- Monteith method to estimate $G_{cw}$, we used a simple flux-gradient method to infer this conductance as follows:

$$G_{CW} = F_{H2O} . \left(\frac{VPD}{P}\right)^{-1} \tag{4}$$

Where, VPD and P are the vapor pressure deficit and atmospheric pressure (both in units of kPa). Estimated conductances and $F_{OCS}$ calculated using both approaches is shown in figure 1. Similar to Wehr et al., (2017), we find that the boundary layer conductance is not limiting at our site, and therefore $G_{sw} \sim G_{cw}$. Consequently, the resulting flux of OCS from the two methods of estimates of Gs are in fact not dissimilar (especially considering the variability around the means shown in Fig. 1b). Therefore, we decided to trust our simple method since it does not depend on theoretical formulations of stability.

[Figure]

*Figure 1. Mean diurnal cycles of boundary layer and canopy conductance to water vapor transport (a), and resulting OCS flux (b).*

 "line 223: You are talking about computing the change in canopy storage, which is a good idea, but why not do that at all times? In general, the flux through the stomata should be equal to the flux past the eddy flux sensor plus the flux into the canopy storage airspace, i.e. Eq. 1 should have a storage term appended to the right hand side (perhaps you used to have one there, as suggested by your reference to the "first term in right hand side of equation (1)" on line 221?). Here you are saying that when the eddy flux term was near zero, you considered the storage flux term. But the storage flux term might be substantial even when the eddy flux term is not near zero."

We have revised this to exclude nighttime data, and periods when the eddy flux is near zero. In tall canopies such as our site, the portion of canopy that is coupled to the overlying atmosphere changes considerably during the day, and parts of the lower canopy are likely to be always decoupled from the upper canopy as well as above canopy air (Pyles et al., 2004). This has obvious consequences on canopy storage and venting of gases such as $CO_2$ and OCS. We have therefore excluded the storage flux entirely from our estimates of $F_{OCS}$. Moreover, change in storage flux leads to a change in mixing ratios (increase during the day), and is implicit in our formulation of $F_{OCS}$. We acknowledge that storage fluxes provide an important portion of the ecosystem exchange of gases such as $CO_2$ (and OCS) that is missed by the eddy flux measurement, but our approach doesn't aim to infer a turbulent flux. Instead, the goal behind this study is to estimate a 'leaf- flux', assuming that the canopy acts like a big-leaf.

"line 236 (Eq. 6): Given the considerations about energy imbalance and the PM equation raised by Wohlfahrt et al., 2009 (Agricultural and Forest Meteorology 149, 1556– 1559) and by Wehr et al., 2017 (Biogeosciences 14, 389–401), it should be stated why this particular form of the PM equation was used (or why the PM equation was used at all instead of just using sensible heat flux measured from the tower). Those papers indicate that retrieved values of stomatal

conductance can be substantially affected by the choice of equation."

We acknowledge the reviewer's concern regarding the use of the Penman-Montieth method to estimate canopy conductance and have now changed the analyses to use the equation presented in eq. 4.

"- line 329: Regarding "declined precipitously with soil moisture", it is a bit hard to tell from the color scale in Fig. 3a, but it looks like the decline in OCS (which matches the decline in Gc), is better correlated with the rise in VPD than with the drop in soil moisture. People often assume that soil moisture drives seasonal patterns in stomatal conductance (and it surely does at some sites), but it is also possible that the seasonal pattern in Gc and F_OCS is explained entirely by VPD (that was the finding for the mesic Harvard Forest site used in the Wehr et al., 2017 paper you cite). It would be interesting to try to disentangle those two water-related drivers here, at least with a simple regression approach."

Yes, we agree with the reviewer on this. However, since we explicitly use VPD to estimate $F_{OCS}$, it would be circular for us to explain variability in FOCS as a function of VPD. The idea behind showing the relationship with soils moisture was a way to link overstory canopy processes, with changes in soil moisture.

"- line 380-1: Did the estimation of ER from the tower include measurements of canopy CO2 storage? If turbulence is low at night, most of the respired CO2 is probably accumulating in airspaces below the eddy flux sensor."

Flux tower estimates of $CO_2$ flux at the site do not incorporate storage (as computed by a profile). This is in part due to large horizontal advective losses that we are unable to estimate (Sonia Wharton, *pers. comm*) within the tall old-growth canopy. This is another reason why we chose to ignore storage estimates of OCS from this analysis.

"- line 427-9: I don't see how this inference is connected to the preceding observations, and I don't see any evidence in the manuscript that soil moisture (as opposed to VPD) is limiting gas exchange."

We have changed the language in the manuscript. We also provide a simple linear regression that quantifies the relationship of $F_{OCS}$ with soil moisture (Fig. 3b).

"- line 452-5 and 483-4: These inferences are flawed. It is not true that "canopy scale stomatal conductance during these events is dramatically reduced". Figure 6 shows that Gc was not reduced at all during the first heat wave, and was not reduced until the end of the second heat wave, at which point the water flux also dropped. During the third heat wave, Gc was reduced,

but the water flux did not increase. Even more importantly, Gc was estimated based on the assumption that the water flux was exclusively transpiration, so it makes no sense to say that the behavior of Gc implies the increased water flux was not transpiration. If the approach used to calculate Gc is valid, then the increased water flux was indeed due to increased transpiration, on account of an increased VPDL."

We agree with the reviewer that higher water flux is likely due to increased transpiration under high VPD, and have changed the language to reflect this (lines 425-432).

References:

Commane, R., Meredith, L. K., Baker, I. T., Berry, J. A., Munger, J. W., Montzka, S. A., Templer, P. H., Juice, S. M., Zahniser, M. S. and Wofsy, S. C.: Seasonal fluxes of carbonyl sulfide in a midlatitude forest, Proc. Natl. Acad. Sci., 112(46), 14162–14167, doi:10.1073/pnas.1504131112, 2015.

Fares, S., Weber, R., Park, J. H., Gentner, D., Karlik, J. and Goldstein, A. H.: Ozone deposition to an orange orchard: Partitioning between stomatal and non-stomatal sinks, Environ. Pollut., 169, 258–266, doi:10.1016/j.envpol.2012.01.030, 2012.

Foken, T.: 50 years of the Monin-Obukhov similarity theory, Boundary-Layer Meteorol., 119(3), 431–447, doi:10.1007/s10546-006-9048-6, 2006.

Pyles, R. D., Paw U, K. T. and Falk, M.: Directional wind shear within an old-growth temperate rainforest: Observations and model results, Agric. For. Meteorol., 125(1–2), 19–31, doi:10.1016/j.agrformet.2004.03.007, 2004.

Seibt, U., Kesselmeier, J., Sandoval-Soto, L., Kuhn, U. and Berry, J. A.: A kinetic analysis of leaf uptake of COS and its relation to transpiration, photosynthesis and carbon isotope fractionation, Biogeosciences, 7(1), 333–341, doi:10.5194/bg-7-333-2010, 2010.

Su, Z., Schmugge, T., Kustas, W. P. and Massman, W. J.: An Evaluation of Two Models for Estimation of the Roughness Height for Heat Transfer between the Land Surface and the Atmosphere, J. Appl. Meteorol., 40(11), 1933–1951, doi:10.1175/1520-0450(2001)040<1933:AEOTMF>2.0.CO;2, 2001.

Wehr, R., Commane, R., Munger, J. W., Barry Mcmanus, J., Nelson, D. D., Zahniser, M. S., Saleska, S. R. and Wofsy, S. C.: Dynamics of canopy stomatal conductance, transpiration, and evaporation in a temperate deciduous forest, validated by carbonyl sulfide uptake, Biogeosciences, 14(2), 389–401, doi:10.5194/bg-14-389-2017, 2017.

Wohlfahrt, G., Brilli, F., Hörtnagl, L., Xu, X., Bingemer, H., Hansel, A. and Loreto, F.: Carbonyl sulfide (COS) as a tracer for canopy photosynthesis, transpiration and stomatal conductance: Potential and limitations, Plant, Cell Environ., 35(4), 657–667, doi:10.1111/j.1365-3040.2011.02451.x, 2012.

---

## Author Response (AR2)

To the associate editor,

Regarding the submitted manuscript (bg-2018-85), this document contains the following:

1. A point by point response to the reviewer's comments
2. A version of the manuscript highlighting all the changes from the previous version.

Finally, here is a brief list of relevant changes.

1. Storage fluxes have been incorporated into the estimation of OCS flux.
2. Inferences about soil moisture have been changed in accordance with reviewer comments.
3. Box plots have been added to Fig. 7 to help visualize the difference between fluxes observed during heat wave events.

On behalf of all the co-authors, I look forward to hearing from you.

Sincerely,
Bharat Rastogi
October, 2018

GENERAL COMMENTS

The authors have done a nice job improving this manuscript, which is more clear and reads well. I have just a few concerns remaining, noted below. With these addressed, I think the paper will make an important contribution to the literature.

We thank the reviewer for providing another round of detailed and valuable comments to help us improve the manuscript. Please see our responses to each comment below.

SPECIFIC COMMENTS

(1) Regarding the OCS flux estimation (Section 2.8)…

First, in the context of flux-gradient theory, g is the conventional symbol for conductance and I think it should not be used for "gradient".
We agree and have changed the symbol for gradient to delta ($\Delta$).

Second, I would spell out the theoretical idea a little more accurately in Section 2.8, as it differs significantly from the Commane et al 2015 paper that is cited as its basis. In Commane et al 2015, the flux-gradient approach was applied to turbulent flux; the assumption was that the turbulent conductance (g = F/gradient) for all gases was the same (no normalization for relative diffusivities necessary because turbulent flux is not diffusive), so that the gradients of the two gases (along a wholly above-canopy turbulent path that passed no sources or sinks) and the flux of one could be used to calculate the flux of the other. Here, instead, your gradient is between the inside of the leaf and the canopy-top atmosphere, which is a path that is mostly diffusive (stomata and, partially, the leaf boundary layer) but partly turbulent (the within-canopy airspace and, partially, the leaf boundary layer), and which may include unaccounted-for sources/sinks in the form of storage flux (i.e. changes in the canopy airspace concentrations) and lateral advection. By normalizing by the diffusivity ratio (i.e. treating the whole path as diffusive) and by neglecting storage and advection, you are effectively taking your canopy-top measurements to represent leaf surface measurements and your eddy flux to represent flux through the stomata. That will definitely invoke some error (possibly bias) and may or may not be a sufficient approximation, so I think the paper should discuss these assumptions and approximations that you are making, how they differ from the cited works, and the resulting uncertainty in the results (ideally quantitatively).

In your response to the first-round review, you argued: "In tall canopies such as our site, the portion of canopy that is coupled to the overlying atmosphere changes considerably during the day, and parts of the lower canopy are likely to be always decoupled from the upper canopy as well as above canopy air (Pyles et al., 2004). This has obvious consequences on canopy storage and venting of gases such as CO2 and OCS." I agree, and this is exactly why taking canopy-top concentrations and fluxes to represent leaf-surface concentrations and fluxes will invoke error. It is also exactly why storage should be accounted for (not neglected, as you argued). If a significant portion of the flux through the stomata is going into (or coming from) changes in canopy concentration, then your flux gradient equations will become significantly inaccurate.

We thank you for this careful and detailed explanation regarding the turbulent and diffusive modes of resistance to flow of gases between the bulk atmosphere and inside the leaves. We showed in our earlier response that at this needle-leaf forest, stomatal (i.e. diffusive) resistance was by far the chief resistance to flow. You are absolutely correct about missing sources/sinks by not accounting for storage fluxes. However, there is considerable uncertainty in the estimation of storage flux, related to the averaging time and vertical resolution of the storage profile (Yang et al., 2007) as well as horizontal resolution (de Araújo et al., 2010; Nicolini et al., 2018). This is especially true at our site, where storage estimates can vary tremendously depending on how this term is estimated. Three different estimates of storage flux are shown (Fig. 1), as estimated from profile measurements. The first is using discrete measurements at the reference height ($S_{70}$), the second is by using measurements along the entire profile ($S_{70-1}$), and finally one using only the two heights at the canopy top ($S_{70-60}$). Importantly, over the course of the average day, each flux cumulatively sums up to zero. At Wind River, storage fluxes for $CO_2$ have been estimated using a discrete measurement only at the reference height (Falk et al., 2008). However, this results in a flux that is not significantly different from 0, which we know is incorrect. If instead, the entire profile is used, the mid-morning storage flux exceeds inferred GPP and is over twice the estimated turbulent flux. This is also likely incorrect, given that wind speed is usually high at 70m and the needle-leaf canopy top is well coupled to the overlying atmosphere. However, the understory at the site is decoupled from the overstory at all times of day, and movement of air in the sub-canopy is controlled by topographically generated (mountain/valley) katabatic flows; this is understandable due to the tall and dense canopy. Thus, turbulent fluxes measured by EC are principally influenced by the upper canopy layers. Incorporating measurements of the decoupled understory can account for large errors in flux estimation (e.g. Jocher et al., 2018). Given the within-canopy decoupling we use estimated storage flux based only on the top two measurement heights (i.e., use the 70m and 60m inlets), allowing us to incorporate the effect of change in storage on the estimation of an ecosystem OCS flux that is predominantly from the upper canopy .

[Figure]

*Figure 1. Mean diurnal cycles of OCS and $CO_2$ storage flux estimated using three different methods (a-b), and the storage flux estimated using the entire profile ($S_{70-1}$) compared to the NEE turbulent flux and the derived GPP (c-d).*

(2) Regarding the response to heat waves (Section 3.5 and lines 462-464)…

This draft is better in that unsupported claims have been removed; however, the trade-off is that there doesn't seem to be a proper conclusion from the heat wave analysis. The only one offered is in the Conclusion section, which says that "sequential heatwaves lead to suppression in stomatal gas exchange of all three fluxes" — but that is not quite true, as F_H2O is enhanced during the heatwaves. Moreover, the reduction of F_OCS is not so clear for the first two heat waves in Fig. 7, probably just due to the noise. Perhaps a regression plot or even just a simple statistic on heat-wave vs not-heat-wave F_OCS would help support the point that F_OCS, in keeping with GPP and Gc, is suppressed during heat waves.

We have added boxplots that show that during these heatwaves OCS uptake and canopy conductance are reduced, NEE is more positive, while $F_{H2O}$ fluxes are not significantly different. We have also added means for $F_{OCS}$ in the text (line 453) that show the difference in these fluxes, and changed a line in the conclusion section (line 492) to say while $F_{CO2}$ and $F_{OCS}$ are suppressed, $F_{H2O}$ is not.

(3) Regarding the influence of soil moisture…

I still do not see how the data support the inference that soil moisture (rather than a combination of VPD, temperature, and light) is driving seasonal changes in gas exchange, and although this draft backs away from strong statements about the soil moisture influence, it still suggests that soil moisture is a key driver in some places (e.g. abstract lines 28-29: "OCS fluxes tracked changes in soil moisture", and lines 411-413: "declining soil moisture likely limits gas exchange as the summer progresses, even as canopy conductance can be reasonably high under overcast skies"). Where soil moisture influence is suggested (and it is certainly a possibility that should be mentioned), it should be on par with other plausible explanations. It is probably also a good idea in general to note somewhere in the manuscript that it is not possible using the present data to tell the soil moisture influence apart from other factors. You attempt to disentangle the drivers in the discussion surrounding Fig 6, but I don't follow the logic. I don't see the Gc response being any more similar across time periods than the OCS or CO2 responses are, despite the text's claim. Nor do I see why such an observation, if it were true, would implicate soil moisture as opposed to temperature or seasonal light levels.

We agree with the reviewer and have now removed soil moisture from the abstract (lines 29-29), and have changed lines 411-413 to say: "This indicates that declining soil moisture (Fig. 3b-c) potentially limits gas exchange as the summer progresses, even as canopy conductance can be reasonably high under overcast skies. It is important to note that in the absence of concurrent leaf and root water potential measurements, it is not possible to attribute reduction in gas exchange due to declining soil moisture."

TECHNICAL CORRECTIONS

- line 235: Eq. 6 seems inverted. The correct equation is flux = conductance x gradient => conductance = flux/gradient
Changed appropriately.
- lines 278-280: Fig 2 does not show the nighttime OCS flux.
We have removed these form the text
- Fig 2d: color scale should have a label (PAR).
Included in the current version.
- line 334: do you mean rainy days?
No. We have changed this to say decreased (line 343).
- line 376: "diffused: total" should be "diffuse:total" (no d and no space)
Changed appropriately.

References

[revised manuscript text omitted]